# LongRoPE2: Near-Lossless LLM Context Window Scaling

**Ning Shang** [* 1]  **Li Lyna Zhang** [* 1 †]  **Siyuan Wang** [2 1]  **Gaokai Zhang** [3 1]  **Gilsinia Lopez** [1]
**Fan Yang** [1]  **Weizhu Chen** [1]  **Mao Yang** [1]

## Abstract

LongRoPE2 is a novel approach that extends the *effective* context window of pre-trained large language models (LLMs) to the target length, while preserving the performance on the original shorter context window. This is achieved by three contributions: **(1)** a hypothesis that insufficient training in higher RoPE dimensions contributes to the persistent out-of-distribution (OOD) issues observed in existing methods; **(2)** an effective RoPE rescaling algorithm that adopts evolutionary search guided by "needle-driven" perplexity to address the insufficient training problem; **(3)** a mixed context window training approach that fine-tunes model weights to adopt rescaled RoPE for long-context sequences while preserving the short-context performance with the original RoPE. Extensive experiments on LLaMA3-8B and Phi3-mini-3.8B across various benchmarks validate the hypothesis and demonstrate the effectiveness of LongRoPE2. Remarkably, LongRoPE2 extends LLaMA3-8B to achieve a 128K *effective* context length while retaining over 98.5% of short-context performance, using only 10B tokens – 80x fewer than Meta's approach, which fails to reach the target effective context length.

## 1  Introduction

A long context window has become an essential feature of Large Language Models (LLMs) (Achiam et al., 2023; Dubey et al., 2024; Abdin et al., 2024; Zhu et al., 2024; Team, 2024). For instance, a 128k context window is now standard in recent LLMs like GPT-4o and LLaMA3.1. Context window extension is achieved through mid-training after pre-training, where the rotary positional embeddings

(RoPE) (Su et al., 2021) are rescaled to fit the expanded context. The model weights are then fine-tuned using long-sequence data to adapt to the rescaled RoPE.

Extending the context window of a pre-trained LLM requires addressing the out-of-distribution (OOD) issue in rotary positional embeddings (RoPE). In RoPE, higher-dimensional RoPE embeddings produce OOD values at extended token positions due to incomplete rotation periods within the original context window (Liu et al., 2023; Han et al., 2023; Men et al., 2024a). To mitigate this, RoPE rescaling remaps these OOD values into the in-distribution range learned during pre-training. Various methods, such as YaRN (Peng et al., 2023), NTK (LocalLLaMA, 2023), and LongRoPE (Ding et al., 2024), have been proposed to determine appropriate rescaling factors.

Despite attempts to mitigate the OOD issue with RoPE rescaling, context window extension still encounters two major challenges. First, rescaling factors derived from previous methods often fall short of achieving the *effective* target context length. For example, LLaMA3.1 adopts YaRN to extend its context window to 128k; however, its performance on RULER (Hsieh et al., 2024), a benchmark designed to evaluate LLMs' long-context processing capability, deteriorates significantly when going beyond 64k (Fig. 1). Second, existing approaches to extending an LLM's context window usually lead to a noticeable performance degradation on tasks for the original short context window. As shown in Fig. 2(c), extending Phi3-mini (Abdin et al., 2024) to 128k results in MMLU score drops of 7.56, 4.34, and 3.52 points for YaRN, NTK, and LongRoPE, respectively. Restoring short-context performance typically requires costly mid-training strategies, such as multi-stage progressive extension (Dubey et al., 2024) and pre-training data replay (Hu et al., 2024b), which increase both training costs (e.g., 800B tokens for LLaMA3.1) and system complexity.

This paper introduces LongRoPE2, a novel approach for context extension that enables LLMs to achieve an effective long context window while preserving short-context performance. Our analysis reveals that lower RoPE dimensions are sufficiently trained, whereas higher dimensions – critical for long-context processing – receive inadequate training. This results in shorter effective RoPE rotation

---

[*]Equal contribution  [1]Microsoft  [2]Shanghai Jiao Tong University  [3]Zhejiang University; Siyuan Wang and Gaokai Zhang did this work during the internship at MSRA. Correspondence to: Li Lyna Zhang <lzhani@microsoft.com>.

*Proceedings of the $42^{nd}$ International Conference on Machine Learning*, Vancouver, Canada. PMLR 267, 2025. Copyright 2025 by the author(s).

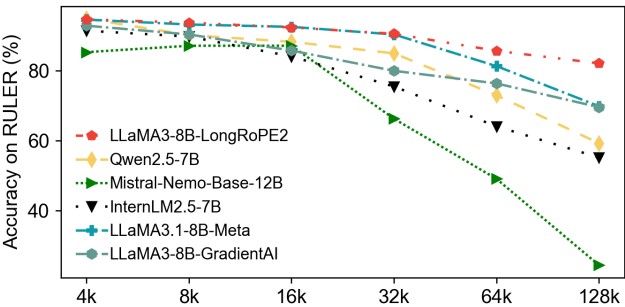

*Figure 1.* LongRoPE2-extended LLaMA3-8B achieves the best performance at a 128k context length among ~10B models.

ranges within the pre-trained context length. We hypothesize that this undertraining in higher dimensions is the root cause of their extended rotation periods longer than their theoretical predictions. Consequently, the critical dimensions shift earlier, leaving existing rescaling methods unable to fully address OOD issues across all dimensions. This hypothesis also explains the empirical observations showing that RoPE requires scaling factors larger than analytically derived values in the higher dimensions for better long-context performance (Gao et al., 2024; Meta, 2024).

Building on this hypothesis, LongRoPE2 adopts a simple yet effective RoPE rescaling algorithm to fully address the OOD issues across all RoPE dimensions. It leverages evolutionary search to identify the true critical RoPE dimensions and optimal rescaling factors, guided by a more effective "needle-driven" perplexity (PPL) evaluation. Unlike conventional PPL, which averages across all tokens, LongRoPE2 focuses exclusively on "needles" – specific answer tokens within long documents that require deep contextual understanding. This ensures accurate evaluation of long-context performance. The search determines the true critical dimensions and rescaling factors for higher OOD dimensions, while NTK scaling is applied to the well-trained lower dimensions. The rescaling factors yielding the lowest PPL are selected as the final solution.

To preserve the original short-context performance, LongRoPE2 incorporates mixed context window training, which simultaneously trains a pre-trained context window with the original RoPE and a long-context window with rescaled RoPE. The long-context window is trained by adapting model weights to the rescaled RoPE for long documents packed to the target length. Concurrently, the short-context window is trained on short documents, also packed to the same target length, using an attention mask to prevent cross-document attention. At inference, original RoPE is used if the input is within the short context; otherwise, rescaled RoPE is applied. This method optimizes long-context performance without sacrificing short-context performance.

Extensive experiments across various LLM sizes and challenging benchmarks validate our hypothesis and demon-strate the effectiveness of LongRoPE2. For Phi3-mini-3.8B and LLaMA3-8B, our rescaling factors shift the theoretical critical dimension from 31 to 25 and from 35 to 30, respectively. By fully resolving RoPE OOD issues, LongRoPE2-extended Phi3-mini-3.8B and LLaMA3-8B achieve an effective 128k context window, significantly outperforming baselines on both synthetic and real-world long-context benchmarks. Moreover, with mixed context window training, LongRoPE2 is the only RoPE rescaling method that can retain over 97% of the original short-context performance on standard tasks. Remarkably, LongRoPE2-extended LLaMA3-8B-128k surpasses Meta's LLaMA3.1-8B-128k in long-context performance while maintaining comparable short-context accuracy, all achieved with just 10B training tokens—80× fewer than Meta's 800B tokens.

## 2 Context Window Extension and Challenges

### 2.1 Preliminary

**Rotary Position Embedding (RoPE)**. Transformer models require explicit positional information, often in the form of position embedding, to represent the order of input tokens. Our work builds on the Rotary Position Embedding (Su et al., 2021), which is widely used in modern LLMs. Let $m \in [0, c)$ be a position index and $\mathbf{x_1}, ..., \mathbf{x_L} \in \mathbb{R}^{|d|}$ a sequence of vectors, where $d$ is the attention head dimension. Using RoPE, the self-attention first incorporates position information to the word embeddings and transforms them into query and key representations:

$$\mathbf{q}_m = f_q(\mathbf{x}_m, m); \quad f_q(\mathbf{x}_m, m) = e^{im\theta}\mathbf{W}_q\mathbf{x}_m \quad (1)$$

$$\mathbf{k}_n = f_k(\mathbf{x}_n, n); \quad f_k(\mathbf{x}_n, n) = e^{in\theta}\mathbf{W}_k\mathbf{x}_n \quad (2)$$

where $i = \sqrt{-1}$ is the imaginary unit. $\mathbf{W}_q, \mathbf{W}_k \in \mathbb{R}^{|d| \times |d|}$ are projection matrices. Attention weights are computed as:

$$softmax(\frac{\mathbf{q}_m^T \mathbf{k}_n}{\sqrt{d}}) \quad (3)$$

where $\mathbf{q}_m$, $\mathbf{k}_n$ are column vectors, and $\mathbf{q}_m^T \mathbf{k}_n$ is their Euclidean inner product. Let $\text{Re}[\cdot]$ denote the real part of a complex number, the inner product $\mathbf{q}^T\mathbf{k}$ becomes:

$$\mathbf{q}_m^T \mathbf{k}_n = \text{Re}\left[(\mathbf{W}_q\mathbf{x}_m)(\mathbf{W}_k\mathbf{x}_n)^* e^{i(m-n)\theta}\right] \quad (4)$$

where $(\mathbf{W}_k\mathbf{x}_n)^*$ is the complex conjugate of $(\mathbf{W}_k\mathbf{x}_n)$. With RoPE, attention becomes a function only dependent on the relative position $m - n$ between tokens, rather than their absolute positions. By applying Euler's formula, $e^{in\theta}$ can be expressed as trigonometric functions. Then, RoPE encodings can be further written as a block diagonal matrix with entries of the form:

$$f_{q,k}(n)_i = \begin{pmatrix} \cos n\theta_i & -\sin n\theta_i \\ \sin n\theta_i & \cos n\theta_i \end{pmatrix}; \theta_i = \theta_{base}^{-2i/d} \quad (5)$$

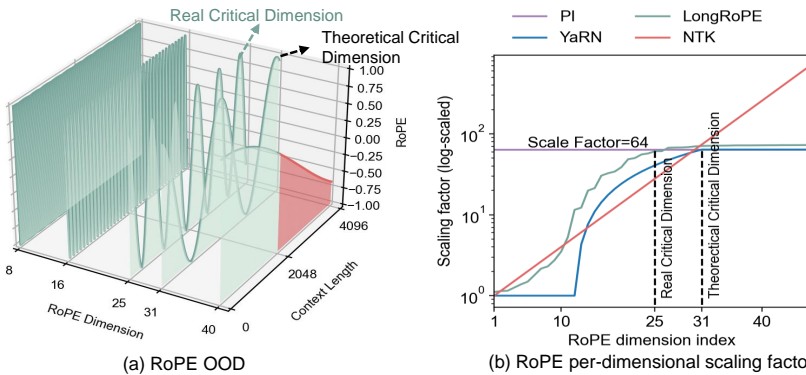

<table>
<tr><th></th><th>RULER 128k
(long context)</th><th>MMLU
(short regular tasks)</th></tr>
<tr><td>Phi3-Mini</td><td>-</td><td>70.78</td></tr>
<tr><td>YaRN</td><td>39.37</td><td>63.22</td></tr>
<tr><td>NTK</td><td>49.37</td><td>66.43</td></tr>
<tr><td>LongRoPE</td><td>**53.71**</td><td>**67.26**</td></tr>
</table>

| | RULER 128k (long context) | MMLU (short regular tasks) |
|---|---|---|
| Phi3-Mini | - | 70.78 |
| YaRN | 39.37 | 63.22 |
| NTK | 49.37 | 66.43 |
| LongRoPE | **53.71** | **67.26** |

(a) RoPE OOD  (b) RoPE per-dimensional scaling factor  (c) Performance of Phi3-mini extended to 128k using different extension methods

*Figure 2.* (a) RoPE OOD (red area) when extending context length from 2k to 4k. (b) Per-dimensional RoPE rescaling factor from different approaches for extending Phi3-mini from 2k to 128k, all aligning with RoPE OOD theory. (c) Performance of Phi3-mini-128k after fine-tuning. Existing methods fail to achieve an effective 128k context length and show noticeable short-context performance drop.

where $\theta_i$ is the per-dimensional rotation angle for $i = 0, 1, ..., d/2 - 1$. $\theta_{base}$ is a predefined RoPE base value, typically set to 10000 in pre-training.

**RoPE Per-Dimensional Period**. Due to the periodicity of $cosine$ and $sine$ functions, RoPE is a periodic function. Specifically, for the $i^{th}$ RoPE dimension, the corresponding period length $T_i$ can be calculated as follows:

$$T_i = \frac{2\pi}{\theta_i} \qquad (6)$$

The period length of each dimension is directly determined by its rotary angle $\theta_i$. As shown in Fig. 2(a), with a fixed $\theta_{base} = 10000$, $\theta_i$ decreases as the dimensional index $i$ increases, leading to longer periods in higher RoPE dimensions. In typical cases, the periods in higher RoPE dimensions often exceeds the pre-trained context window size, leading to incomplete periods. For instance, in Phi3-mini, the pre-trained context window size is 2048, while the period length of the highest dimension (i.e., the $48^{th}$ $cosine$ dimension) is 51861, covering less than $4\%$ of a full period.

## 2.2 RoPE Rescaling Theory

Despite its effectiveness, RoPE, like other position encodings, faces challenges in context length extrapolation. In particular, when input sequence length exceeds the predefined context window, the perplexity can shoot up to levels comparable to completely untrained models (i.e., $> 10^3$).

**RoPE OOD.** Direct length extrapolation fails because longer sequences introduce untrained token positions, leading to out-of-distribution (OOD) positional values in RoPE. As shown in Fig. 2(a), the periods in high RoPE dimensions exceed the original context window size $L_{train}$. Consequently, for these dimensions, the model does not see a full rotation period during pre-training, resulting in new untrained RoPE values at extended token positions. For instance, in Fig. 2(a), the $40^{th}$ $cosine$ dimension does not complete a full period within the pre-trained length $L_{train}$=2k.

When directly extrapolated to 4k, the $cosine$ values between 2k and 4k fall outside the pre-trained range, becoming OOD RoPE values (highlighted in red).

***Theoretical Critical RoPE dimension***. In contrast to higher RoPE dimensions, lower dimensions (e.g., $8^{th}$ and $16^{th}$ dimension in Fig. 2(a)) have seen many full periods during pretraining. As a result, there exists a **theoretical critical dimension** (TCD) $d_{tcd}$ that divides RoPE dimensions into two groups: one with multiple full periods within the pre-trained length $L_{train}$ (i.e., $T_i < L_{train}, i < d_{tcd}$) and another with incomplete periods (i.e., $T_i \geq L_{train}, i \geq d_{tcd}$). Following (Liu et al., 2023), the critical dimension can be computed as:

$$d_{tcd} = 2\lceil \frac{d}{2} \log_{\theta_{base}} \frac{L_{train}}{2\pi} \rceil \qquad (7)$$

As shown in Fig.2(a), for Phi3-mini(Abdin et al., 2024) with $d$=96, a base $\theta_{base}$=10000, and $L_{train} = 2048$, the critical dimension is 62, corresponding to the $31^{st}$ $cosine$ dimension. Unless otherwise specified, we focus on the $cosine$ dimensions of RoPE (i.e., $i = 0, 1, ..., d/2 - 1$) for simplicity.

***RoPE OOD theory***. To address the RoPE OOD issue in long-context extension, a straightforward approach is to rescale the per-dimensional rotation angle $\theta_i$ and ensure higher RoPE-OOD dimensions remain within the pretrained RoPE range. This forms the widely accepted RoPE OOD theory (Liu et al., 2023; Chen et al., 2023; Men et al., 2024a).

Formally, let the target context window size be $L$ and $\lambda_i$ be the rescaling factor for the $i^{th}$ RoPE dimension. The rescaled per-dimensional rotation angle $\hat{\theta}_i$ is then given by:

$$\hat{\theta}_i = \frac{1}{\lambda_i \times \theta_{base}^{2i/d}} \qquad (8)$$

To avoid OOD, the new rescaled periods of higher RoPE dimensions ($\hat{T}_i, i > d_{cd}$) must remain within the pretrained

range, leading to the following constraint:

$$\frac{L}{\hat{T}_i} \leq \frac{L_{\text{train}}}{T_i}; \rightarrow \frac{L\hat{\theta}_i}{2\pi} \leq \frac{L_{\text{train}}\theta_i}{2\pi}; \quad \text{for} \quad i \geq d_{\text{tcd}} \quad (9)$$

$$\lambda_i \geq \frac{L}{L_{\text{train}}}; \quad \text{for} \quad i \geq d_{\text{tcd}} \quad (10)$$

Specifically, $\frac{L}{L_{\text{train}}}$ is the context window extension ratio. The RoPE OOD theory establishes this ratio as the lower bound for scaling factors in higher RoPE dimensions beyond $d_{tcd}$.

### 2.3 Review of Prior RoPE Rescaling Approaches

Building on the RoPE OOD theory, various RoPE rescaling methods have been proposed for LLM context window extension (Chen et al., 2023; Han et al., 2023; Men et al., 2024b; Yang et al., 2024b). Prominent approaches, including PI, NTK, YaRN and LongRoPE, have been widely adopted to enable long context in open-source LLMs (Yang et al., 2024a; Dubey et al., 2024; Abdin et al., 2024).

**PI** introduces linear positional interpolation, where all the RoPE dimensions use the same scale factor of $\lambda_i = \frac{L}{L_{\text{train}}}$. Despite its simplicity, this uniform scaling "crowds" the positional information, making it difficult for the model to distinguish closely positioned tokens.

**NTK** $\theta$ **Scaling** approaches RoPE from an information encoding perspective, applying the Neural Tangle Kernel (NTK) theory (Jacot et al., 2018; Tancik et al., 2020). The core idea is that neural networks are difficult to learn high-frequency features (low RoPE dimensions), and large scaling factor can affect these high-frequency positional information, leading to the loss of crucial details needed to differentiate similar closely positioned tokens.

As a result, NTK-based methods suggest increasing the original RoPE base value $\theta_{base}$ to a larger base $\theta_{ntk}$. Several methods (LocalLLaMA, 2023; Men et al., 2024b; Liu et al., 2023) have been proposed to determine this new base value. However, some fail to align with RoPE OOD theory. For instance, (LocalLLaMA, 2023) use $\lambda_i = s^{2i/(d-2)}$, leading to insufficient interpolation and increased PPL before the target length. The approach in (Liu et al., 2023), which calculates $\theta_{ntk}$ based on the theoretical critical dimension, is the most widely adopted NTK-based method. Specifically, $\theta_{ntk} \geq \theta^{\log_{\frac{L_{\text{train}}}{2\pi}} \frac{L}{2\pi}}$, yielding $\lambda_i \geq \frac{L}{L_{\text{train}}}(i > d_{tcd})$. Unless stated otherwise, "NTK" in this work refers to this approach.

**YaRN** divides RoPE dimensions into three groups as shown in Fig. 2(b). For lower dimensions with high frequencies, YaRN proposes no interpolation, setting $\lambda_i = 1$ to better preserve high-frequency positional information compared to NTK. For high dimensions, YaRN adopt PI and set $\lambda_i = \frac{L}{L_{\text{train}}}$. For dimensions that fall in-between use a linearly increasing scale factor.

**LongRoPE**. Unlike other extension methods relying on theoretical analysis, LongRoPE employs a PPL-guided evolutionary search to find the per-dimensional scale factor $\lambda_i$. To leverage NTK theory, it enforces a monotonically non-decreasing scaling factor constraint during the search.

### 2.4 Challenges

**RoPE OOD theory are insufficient**. Fig. 2(b) compares scale factor distributions for extending Phi3-mini from 2k to 128k. NTK, YaRN and LongRoPE all align the RoPE OOD with $\lambda_i \geq 64$ for $i > d_{tcd}$, but yielding varied performance (Fig. 2(c)). NTK and LongRoPE outperforms YaRN on both short- and long-context tasks. We highlight two observations: **(1)** The theoretical lower bound, $\frac{L}{L_{\text{train}}}$, is often suboptimal. Beyond dimension $d_{tcd} = 31$, YaRN strictly adheres to this bound ($\frac{L}{L_{\text{train}}}$=64), but NTK and LongRoPE use larger values to achieve much better performance. **(2)** Beyond $d_{tcd}$, larger scale factors don't always improve long-context performance. For example, in dimensions 31-48, NTK uses much larger scale factors than LongRoPE, yet LongRoPE achieves better performance. These findings align with prior works (Meta, 2024; Men et al., 2024a; Wang et al., 2024), where *marginally larger* scale factors than the extension ratio empirically improve performance.

This raises the fundamental question: *In RoPE OOD theory, if RoPE periods beyond critical dimension can address OOD with $\lambda_i = \frac{L}{L_{train}}$, why do slightly larger scaling factors lead to better performance?*

**Short performance drop**. A persistent challenge in long context extension is performance degradation on original short window, which poses a significant obstacle in practical LLM development. A common solution is progressively extension using large-scale training data (Dubey et al., 2024; Hu et al., 2024b). For example, LLaMA3.1 (Dubey et al., 2024) adopts a *SIX-stage* extension process requiring 800B tokens to extend from 8k to 128k, greatly increasing training complexity and costs. Though LongRoPE introduces a training-free short scaling factor, it fails to fully address the performance drop (Figure 2(c)). As a result, bridging this gap remains an unresolved challenge.

## 3 LongRoPE2 Methodology

### 3.1 New RoPE OOD Hypothesis

**The empirical RoPE periods in higher dimensions are longer than theoretical values, limiting current methods to fully address RoPE OOD**. In Sec. 2, we observe that RoPE scale factors slightly exceeding the theoretical lower bound beyond the critical dimension $d_{tcd}$ yield improved long-context performance. We attribute this to insufficient training in higher dimensions, which extends rotation periods and reduces the critical dimension index (Fig. 2(a)) relative to the theoretical expectations.

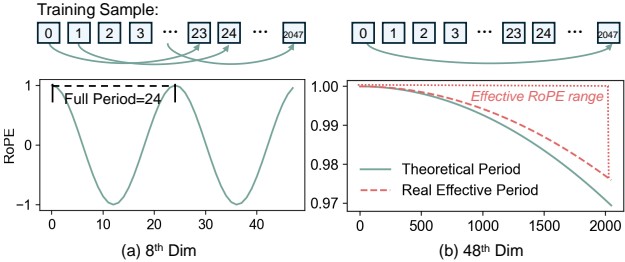

*Figure 3.* Sequence length required to span the theoretical period during Phi3-mini pre-training for different RoPE dimensions. Insufficient training in higher RoPE dimensions leads to shorter effective RoPE ranges and longer actual periods.

As illustrated in Fig. 3(a), lower RoPE dimensions (with shorter periods) receive repeated full-period training cycles within a single corpus. For example, in Phi3-mini, the $8^{th}$ dimension has a short period of 24, requiring only $m - n = 24$ tokens for a full cycle. A 2048-token training sample thus covers this dimension thousands of times, ensuring sufficient training. In contrast, higher RoPE dimensions, with period exceeding the pre-trained context window, receive far less training. For example, the $48^{th}$ dimension spans only $\sim 4\%$ of its *cosine* period within a 2048-token sequence (Fig. 3(b)), resulting in the theoretical incomplete period being covered just once.

A deeper challenge arises after self-attention: these incomplete RoPE periods in high dimensions exhibit reduced effective ranges (Fig. 3(b)), stretching practical period beyond theoretical values. As shown in Eq. 3, RoPE positional information is incorporated via self-attention, where the max relative token distance determines the practical RoPE range. As real-world data rarely contains long-range dependencies (e.g., distances of 2048 tokens), higher RoPE dimensions tend to be under-trained, amplifying period discrepancies.

This under-training in higher RoPE dimensions explains why larger scaling factors improve long-context performance. We formalize this insight as:

**Hypothesis**. *Insufficient training in higher RoPE dimensions extends empirical rotation periods beyond the theoretical $\frac{2\pi}{\theta_i}$. This discrepancy necessitates larger scale factors to mitigate RoPE OOD and lowering the critical dimension index $d_{rcd}$ below its theoretical $d_{tcd}$.*

### 3.2 RoPE Rescaling Factor Search

Since the theoretical RoPE OOD theory cannot fully address OOD issues, we use a search-based approach to identify the practical true critical dimension and optimal rescaled RoPE. Inspired by LongRoPE, we search for scaling factors, apply them to the pre-trained LLM via rescaled RoPE, and compute perplexity (PPL) on fixed samples at a target context length (e.g., 128k). The factors that minimize PPL are chosen for best preserving pre-trained RoPE information while addressing OOD. Given that the approach relies entirely on

---

**Algorithm 1** Initialization with theoretical periods

**Input:** theta base $\theta_{base}$; RoPE dim $d$, pre-trained context window size $L_{train}$, target length $L$; theoretical critical dimension $d_{tcd}$

1: $P_0 = [0] * 2/d$
2: $d_{tcd}^{10} = \lceil \frac{d}{2} \log_{\theta_{base}} \frac{L_{train}}{2\pi \times 10} \rceil$ {Compute the dim with a theoretical 10 periods.}
   {include smaller indices as candidate $d_{rcd}$}
3: **for** int $d_{rcd} = d_{tcd}^{10}$ to $d_{tcd}$ **do**
4:     s=randint($\frac{L}{L_{train}}, 2 \times \frac{L}{L_{train}}$)
5:     $\lambda[d_{rcd} : \frac{d}{2} - 1] = s$
6:     $\theta_{d_{tcd}^{10}} = \frac{1}{s \times \theta_{base}^{(2 \times d_{tcd}^{10}/d)}}$
7:     $\lambda[0 : d_{rcd}]$= compute rescaling factors using NTK $\theta_{d_{tcd}^{10}}$
8:     add $\lambda$ into $P_0$;
9: **end for**
10: Return $P_0$ ;

---

the search, we introduce two key innovations.

**Synthetic needle data to guide the search**. Naively using PPL-guided search can easily result in suboptimal rescaling factors. First, long sequences often contain irrelevant or low-dependency tokens, reducing the effective maximum token dependency. For instance, predicting the final token in a 128k-token book may not require the context of the first token. Second, standard PPL, by averaging over all token equally, fails to effectively capture the long-context abilities (Hu et al., 2024a; Fang et al., 2024) and can be dominated by irrelevant tokens, obscuring key answer tokens. As a result, the rescaling factors that minimize PPL often fail to achieve the target context window size.

To address this, we introduce a needle-driven PPL evaluation. Instead of using real-world long documents, we synthesize long data with controlled token dependency distances. Inspired by needle retrieval benchmarks for long-context evaluation (Hsieh et al., 2024; Li et al., 2024a), we randomly sample 10 books from the PG19 validation set. At the start of each sample, we insert a "needle" (a specific piece of text as shown in Appendix B), and at the end, we ask the model to retrieve this needle. We then compute the perplexity of only the retrieved needle tokens. The needle-based PPL evaluates how well the model, with the rescaled RoPE, can understand the entire context and retrieve the distant needle.

**Critical dimension-aware scale factor search**. With the synthetic needle-driven PPL evaluation, we run a simple evolutionary search to identify the real critical dimension $d_{rcd}$ and the optimal rescaling factors. For search efficiency, we restrict the search to dimensions $i \geq d_{rcd}$, while applying NTK-aware scaling to lower dimensions ($i < d_{rcd}$) using the adjusted base value derived from $d_{rcd}$.

The search begins by initializing $d_{rcd}$ and rescaling factors, as detailed in Algorithm 1. Based on our hypothesis, smaller indices are considered potential $d_{rcd}$, with candidates ranging from $d_{tcd}^{10}$, where the theoretical RoPE period spans 10 periods in the pre-training window, and $d_{tcd}$. For each can-

**Algorithm 2** Critical dimension aware mutation

---

**Input:** population P; mutation probability $p$; synthetic long data $\mathbf{X}$

1: Top-k = $Update\_Topk$ ( P);
2: SP=$[\frac{L}{L_{\text{train}}}, 2 \times \frac{L}{L_{\text{train}}}]$ {search space}
3: **for** $\lambda$ in Top-k **do**
4: $\quad \lambda_{right} = \lambda[d_{rcd} : \frac{d}{2} - 1]$
5: $\quad \lambda_{right}$=Mutation_with_mono_constraint ($\lambda_{right}$, $p$, SP) {mutate scale factors beyond $\theta_{d_{rcd}}$.}
6: $\quad \lambda[d_{rcd} : \frac{d}{2} - 1] = \lambda_{right}$
7: $\quad \theta_{d_{rcd}} = \frac{1}{\lambda_{right}[0] \times \theta_{base}^{(2 \times d_{rcd}/d)}}$ {update theta base in $\theta_{d_{rcd}}$.}
8: $\quad \lambda[0 : i]$= compute rescaling factors using NTK $\theta_{d_{rcd}}$ {update dims before $\theta_{d_{rcd}}$}
9: $\quad Compute\_PPL$ (LLM, $\lambda$, $\mathbf{X}$); add $\lambda$ into P;
10: **end for**
11: Update P with Top-k; Return the latest population P ;

---

didate, rescaling factors above $\frac{L}{L_{\text{train}}}$ are randomly sampled for dimension $i \geq d_{rcd}$ to address RoPE OOD value, while NTK scaling is applied to dimensions $i < d_{rcd}$.

We iteratively sample and mutate rescaling factors until reaching a population size $N$. Using the needle-driven synthesis method, we generate $L$-token documents and compute PPL for each candidate by applying the rescaling factors to the LLM and evaluating the input $\mathbf{X}$.

The population is updated through standard evolution search. Algorithm 2 shows the mutation process. For each sampled scaling factor, we split RoPE dimensions at $d_{rcd}$. The higher group ($i \geq d_{rcd}$) performs mutation with probability $p$ under the monotonic non-decreasing constraint: $\lambda_i \leq \lambda_{i+1}$. The theta base for $d_{rcd}$ is updated after mutation, and NTK scaling is applied to rescale factors in the lower group.

Fig. 4 shows the final scaling factors identified by LongRoPE2 for Phi3-mini and LLaMA3-8B under a 128k context. The practical critical dimensions ($d_{rcd}$) are shifted earlier to 25 and 30, compared to the theoretical values $d_{tcd}$ of 31 and 35, respectively. The scaling factors for RoPE OOD dimensions are slightly larger than PI/YaRN/LongRoPE and notably smaller than NTK.

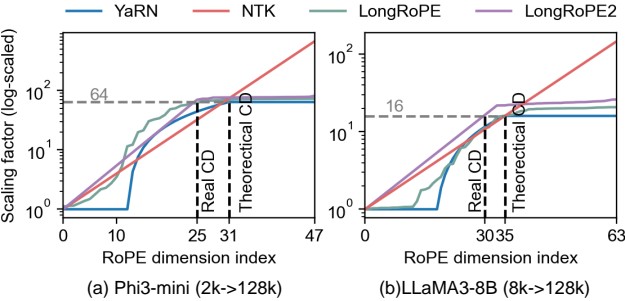

*Figure 4.* Scale factors across different RoPE rescaling approaches.

### 3.3 Mixed Context Window Training

We then apply the optimal rescaling factors to RoPE on the pre-trained LLM, but two critical challenges remains for effective long-context LLM deployment. First, the pre-trained

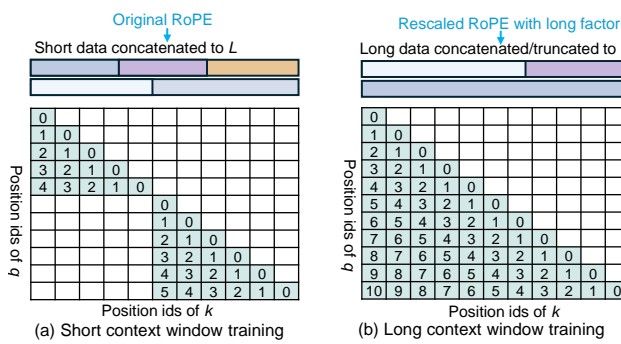

*Figure 5.* Mixed context window training to improve both short and long context capabilities.

*Table 1.* Mid-training data mix.

| | Short Context Window $\leq L_{train}$ | Long Context Window $L_{train}$-100k | 100k-200k |
|---|---|---|---|
| Tokens | 3B | 3B | 4B |

model weights have not been trained with the rescaled RoPE, leading to poor performance on real-world long-context tasks. Second, extending context window size often degrades performance on original short-context tasks (Ding et al., 2024; Hu et al., 2024b), making it challenging to balance long- and short-context capabilities.

To address these challenges, we introduce a novel mixed context window training approach that achieve both long- and short-context superior performance without adding system-level training complexity. Specifically, short-context training reuses the original RoPE and fine-tunes on short sequences, preserving pre-trained performance. Long-context training applies the rescaled RoPE and fine-tunes on long sequences, enabling effective long-context understanding.

Fig. 5 illustrates this process. For a target context window size of $L$=128k, we sample short sequences ($\leq L_{\text{train}}$) and long sequences (8k-200k), chunked into 128k segments with BOS and EOS tokens. For segments labeled as short windows, the original RoPE is used with attention masks to prevent self-attention across different documents as shown in Fig. 5(a). For long-context segments, we apply the rescaled RoPE for full attention within the 128k segments (Fig. 5(b)). This design contrasts with prior "mixed training" methods such as LLaMA-3.1 (Dubey et al., 2024) and LongAlign (Bai et al., 2024), which use a single long RoPE scaling factor and disallow cross-document attention. In contrast, our approach employs dual RoPE scaling and allows cross-document attention for long contexts. More details can be found in Appendix A.

## 4 Experiments

### 4.1 Setup

**Evaluation LLMs and Tasks**. We apply LongRoPE2 to LLaMA3-8B and Phi3-mini (3.8B). Phi3-mini, with its limited capabilities, serves as a rigorous testbed for evaluating

*Table 2.* Comparison with prior SOTA RoPE rescaling methods on RULER Benchmark. We report the average score across 13 tasks.

| Method | 4k | 8k | 16k | 32k | 64k | 128k |
|---|---|---|---|---|---|---|
| Base Model: Phi3-mini (3.8B) | | | | | | |
| YaRN | 85.74 | 78.68 | 75.97 | 65.22 | 52.16 | 39.37 |
| NTK | **91.34** | 87.02 | 80.57 | 72.81 | 61.91 | 49.37 |
| LongRoPE | 88.40 | 83.23 | 79.46 | 71.20 | 64.63 | 53.71 |
| **LongRoPE2** | 90.41 | **87.22** | **83.33** | **76.51** | **65.37** | **58.81** |
| Base Model: LLaMA3-8B | | | | | | |
| YaRN | 91.86 | 87.87 | 84.67 | 68.80 | 62.51 | 49.39 |
| NTK | 94.38 | 92.64 | 91.93 | 87.33 | 79.26 | 73.19 |
| LongRoPE | 94.60 | 92.70 | 91.01 | 86.60 | 81.23 | 73.40 |
| **LongRoPE2** | **94.61** | **93.68** | **92.31** | **90.49** | **85.62** | **82.03** |

RoPE rescaling methods. Performance is evaluated across three dimensions: (1) long-context stress tests, including RULER (Hsieh et al., 2024) and Needle in a Haystack (Kamradt, 2023); (2) real-world long-context benchmarks including LOFT (Lee et al., 2024a), InfiniteBench (Zhang et al., 2024a), and LongBench (Bai et al., 2023). Notably, since our method extends a pre-trained LLM without post-training, we prioritize sub-tasks aligned with completion-based tasks and QA tasks with few-shot examples; (3) standard benchmarks within a 4096-token context.

**Mid-training**. Our method can potentially support million-level context length, but due to resources constraint, we extend the two models to 128k context window and mid-train on 64 A100 GPUs using a 10B-token dataset. Following the per-source upsampling from (Fu et al., 2024), we sample 4.5B, 2.5B, and 2B tokens from RedPajama-v1 (Computer, 2023), RedPajama-v2 (Weber et al., 2024), and StarCoder (Li et al., 2023), covering 8k–200k sequence lengths. For short context windows, we sample 1B tokens from Fineweb-Edu (Lozhkov et al., 2024). Table 1 shows the token distribution by sequence length. We train for 1 epoch with a global batch size of 64. The initial learning rate of 2e-5 with a cosine learning rate scheduler.

**Baselines**. We compare with state-of-the-art RoPE rescaling methods, including YaRN, NTK, and LongRoPE. All baselines use the same mid-training procedure for fairness.

### 4.2 Main Results

We present the main results of LongRoPE2-extended Phi3-mini-3.8B-128k and LLaMA3-8B-128k, comparing them with models using other STOA RoPE rescaling methods.

**Long-context performance on RULER benchmark**. Table 2 compares performance on RULER, which consists of 13 synthetic tasks. Across Phi3-mini-3.8B and LLaMA3-8B, LongRoPE2 consistently outperforms prior methods, achieving superior results across all evaluation lengths within the 128k window. On LLaMA3-8B, LongRoPE2 achieves an effective 128k context window, maintaining a strong score of 82.03 at 128k, while previous methods degrade signif-

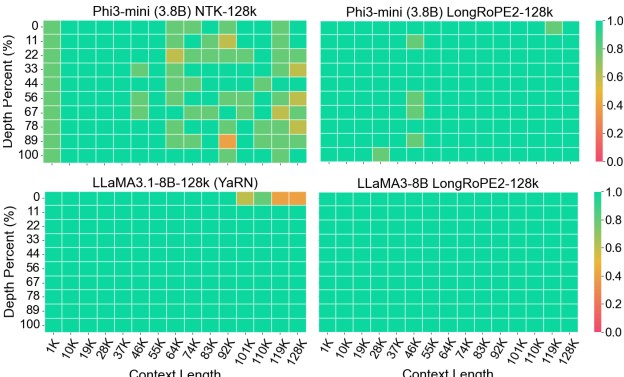

*Figure 6.* LongRoPE2 (right) delivers near-perfect lossless performance in the "Needle in a Haystack" pressure test.

icantly at longer contexts. For example, LongRoPE, the prior best, drops from 81.23 (64k) to 73.40 at 128k. For Phi3-mini-3.8B, LongRoPE2 shows even greater advantages, overcoming the challenges of the smaller model's weaker capabilities. NTK performs well below 32k and declines sharply beyond, while LongRoPE underperforms at shorter contexts. In contrast, LongRoPE2 consistently enhances performance across all lengths. Notably, the 128k average score of 58.81 is skewed by tasks with low scores on smaller LLMs, such as CWE, which achieves only 1% accuracy. Detailed per-task score is available in Appendix A.

**Needle in a Haystack pressure tests**. We evaluate LongRoPE2 using the popular long-context pressure test, Needle in a Haystack, which measures a model's ability to retrieve "needles" from long documents at varying depths. We run 10 times at the same depth and length. As shown in Fig. 6, LongRoPE2 achieves near-perfect accuracy across all evaluation lengths within the 128k context window. In contrast, methods like NTK often fail at longer contexts, and LLaMA3.1-8B extended by YaRN, despite being fine-tuned on 800B tokens, fails beyond 100k. These results highlight LongRoPE2's robust long-context modeling capabilities.

**Long-context performance on real-world benchmarks**. Beyond synthetic tasks, we evaluate real-world benchmarks: LOFT (7 retrieval tasks including argumentative retrieval, fact-checking, web search, multi-hop reasoning QA, etc), InfiniteBench (key-value retrieval and multi-choice QA), and LongBench (in-context learning and code completion). Note that our models are evaluated without post-training, so scores are lower than post-training results. As shown in Table 3, LongRoPE2 consistently improves performance across all benchmarks, demonstrating strong generalization to practical scenarios. In contrast, YaRN and NTK perform notably worse, particularly on the small Phi3-mini-3.8B.

**Standard benchmarks at original context window**. RoPE-based context extension typically sacrifices short-context performance. As Table 4 shows, prior methods like YaRN, NTK, and LongRoPE exhibit notable degradation. For ex-

*Table 3.* Long context performance comparison under different extension methods on real-world benchmarks

| Method | LOFT | | | | | | | | InfiniteBench - LongBench | | | | | | |
|---|---|---|---|---|---|---|---|---|---|---|---|---|---|---|---|
| | Avg. | ArguAna | FEVER | HotPotQA | MS MACRO | NQ | Quora | SciFact | Avg. | KV retrieval | En.MC | TriviaQA | TREC | LCC | RepoBench-P |
| | | | | | | Base model: Phi3-mini (3.8B) | | | | | | | | | |
| YaRN | 5.86 | 4.0 | 4.0 | 0 | 8.0 | 12.0 | 1.0 | 12.0 | 50.96 | 5.8 | 31.44 | 84.35 | 61.00 | **63.98** | 59.23 |
| NTK | 7.57 | 0 | 21.0 | 0 | 6.0 | 13.0 | 4.0 | 9.0 | 52.31 | 5.1 | 37.55 | 84.01 | 65.00 | 62.36 | 59.82 |
| LongRoPE | 21.14 | 5.0 | 64.0 | 3.0 | 17.0 | 35.0 | 8.0 | **16.0** | 50.67 | 5.6 | 35.81 | 86.47 | 62.50 | 55.25 | 58.43 |
| **LongRoPE2** | **23.00** | **5.0** | **70.0** | **4.0** | **19.0** | **39.0** | **10.0** | 14.0 | **55.23** | **12.0** | **42.36** | **87.27** | **67.00** | 62.67 | **60.10** |
| | | | | | | Base model: LLaMA3-8B | | | | | | | | | |
| YaRN | 26.14 | 7.0 | 62.0 | 15.0 | 21.0 | 43.0 | 23.0 | 12.0 | 51.81 | 2.2 | 30.57 | 88.97 | 73.50 | 65.40 | 62.21 |
| NTK | 67.14 | 22.0 | 96.0 | 53.0 | 75.0 | 89.0 | 71.0 | 64.0 | 67.98 | 66.0 | 42.79 | 90.87 | 74.00 | 68.67 | 65.55 |
| LongRoPE | 60.85 | 22.0 | 96.0 | 25.0 | 57.0 | 90.0 | 74.0 | 62.0 | 70.39 | 74.0 | 45.85 | 89.99 | 76.00 | 69.13 | 67.38 |
| **LongRoPE2** | **74.28** | **28.0** | **96.0** | **70.0** | **80.0** | **94.0** | **79.0** | **73.0** | **73.37** | **88.0** | **46.72** | **91.13** | **76.50** | **70.47** | **67.39** |

*Table 4.* Comparison of long-context LLMs with original Phi3-mini and LLaMA3-8B on regular short benchmarks.

| (a) Phi3-mini (3.8B) with 128k context window | | | | | |
|---|---|---|---|---|---|
| Model | Avg. | MMLU | MMLU-Pro | HellaSwag | TruthfulQA | GSM8K |
| Original Phi3-mini (2k) | 63.2 | 70.78 | 41.17 | 77.96 | 47.82 | 78.54 |
| YaRN | 53.6 | 63.22 | 30.95 | 75.27 | 42.19 | 57.39 |
| NTK | 57.3 | 66.43 | 36.09 | 76.92 | 43.34 | 63.99 |
| LongRoPE | 58.5 | 67.26 | 36.28 | 75.73 | 46.26 | 67.17 |
| **LongRoPE2** | **61.7** | **70.04** | **40.30** | **77.07** | **47.61** | **73.62** |
| (b) LLaMA3-8B with 128k context window | | | | | |
| LLaMA3.1-8B | 57.2 | 66.33 | 36.79 | 81.71 | 45.17 | 56.18 |
| Original LLaMA3-8B (8k) | 56.5 | 66.62 | 35.87 | 82.08 | 44.04 | 54.05 |
| YaRN | 52.1 | 62.25 | 31.88 | 81.25 | 42.61 | 42.45 |
| NTK* | 54.0 | 63.84 | 34.14 | 82.11 | 43.45 | 46.92 |
| LongRoPE | 54.6 | 64.69 | 33.74 | **82.14** | 43.65 | 48.90 |
| **LongRoPE2** | **55.7** | **65.01** | **34.61** | 81.69 | **46.17** | **50.80** |

*Table 5.* Ablation study on real critical dimension.

| Method | Regular short tasks | | | RULER | | | | | |
|---|---|---|---|---|---|---|---|---|---|
| | MMLU | MMLU Pro | GSM8K | 4k | 8k | 16k | 32k | 64k | 128k |
| | | | Base Model: Phi3-mini (3.8B) | | | | | | |
| LongRoPE | **70.07** | **40.30** | **73.62** | **90.41** | **87.22** | **83.33** | **76.51** | **65.37** | **58.81** |
| YaRN | **63.22** | **30.95** | **57.39** | 85.74 | **78.68** | **75.97** | 65.22 | 52.16 | 39.37 |
| **YaRN-rcd** | 62.30 | 30.24 | 56.48 | **86.56** | 77.66 | 74.48 | **67.73** | **52.73** | **44.39** |
| NTK | **66.43** | **36.09** | **63.99** | **91.34** | **87.02** | 80.57 | 72.81 | 61.91 | 49.37 |
| **NTK-rcd** | 65.31 | 35.09 | 59.29 | 90.51 | 85.32 | **81.80** | **73.89** | **63.59** | **54.42** |
| | | | Base Model: LLaMA3-8B | | | | | | |
| LongRoPE | **65.01** | **34.61** | **50.80** | **94.61** | **93.68** | **92.31** | **90.49** | **85.62** | **82.03** |
| YaRN | 62.25 | 31.88 | 42.45 | 91.86 | 87.87 | 84.67 | 68.80 | 62.51 | 49.39 |
| **YaRN-rcd** | **64.30** | **33.17** | **50.34** | **94.22** | **92.02** | **89.20** | **82.56** | **76.37** | **71.46** |
| NTK | 63.84 | 34.14 | **46.92** | 94.38 | **92.64** | **91.93** | 87.33 | 79.26 | 73.19 |
| **NTK-rcd** | **64.70** | **34.23** | 45.87 | **94.39** | 92.35 | 91.43 | **88.82** | **83.22** | **77.25** |

*Table 6.* Ablation study on needle-PPL guided search.

| Search Metric | 4k | 8k | 16k | 32k | 64k | 128k |
|---|---|---|---|---|---|---|
| | Base Model: Phi3-mini (3.8B) | | | | | |
| PG19-128k PPL | **91.16** | **87.93** | 83.05 | 75.27 | 62.72 | 50.23 |
| **PG19-Needle 128k PPL (ours)** | 90.41 | 87.22 | **83.33** | **76.51** | **65.37** | **58.81** |
| | Base Model: LLaMA3-8B | | | | | |
| PG19-128k PPL | 94.46 | 93.36 | 91.67 | 90.28 | 84.55 | 78.68 |
| **PG19-Needle 128k PPL (ours)** | **94.61** | **93.68** | **92.31** | **90.49** | **85.62** | **82.03** |

*Table 7.* Ablation study on mixed context window training.

| Method | MMLU | MMLU Pro | GSM8K | 4k | 8k | 16k | 32k | 64k | 128k |
|---|---|---|---|---|---|---|---|---|---|
| | | | Base Model: Phi3 June | | | | | | |
| LongRoPE2 | 70.07 | 40.30 | 73.62 | 90.41 | **86.87** | **83.33** | **76.51** | **65.37** | **58.81** |
| LongRoPE2/ wo. | 66.56 | 34.86 | 64.67 | **90.55** | 85.77 | 81.08 | 73.31 | 63.75 | 56.22 |
| | | | Base Model: LLaMA3-8B | | | | | | |
| LongRoPE2 | 65.01 | 34.61 | **50.80** | 94.61 | **93.68** | **92.31** | **90.49** | **85.62** | **82.03** |
| LongRoPE2/ wo. | 64.57 | 33.83 | 48.37 | **94.67** | 93.15 | 91.24 | 89.38 | 83.53 | 80.18 |

NTK-rcd variants (see Fig. 9 in Appendix A). As shown in Table 5, correcting $d_{rcd}$ improves long-context performance for both methods, revealing the inadequacy of theoretical critical dimensions in fully addressing RoPE OOD issues. However, correcting the critical dimension alone does not ensure optimal results. By further optimizing scaling factors, LongRoPE2 consistently outperforms YaRN-rcd and NTK-rcd on both short- and long-context benchmarks.

**The effectiveness of need-PPL guided search**. LongRoPE2 identifies the true critical dimension and scaling factors through a needle-PPL-guided evolutionary search, which minimizes interference from irrelevant tokens to effectively capture the rescaled RoPE's long-context capabilities. To validate its effectiveness, we use 10 pure PG19 documents as a baseline, identical to those used for generating our needle-data, applying the same search and mid-training process. Table 6 compares the RULER scores for Phi3-mini-3.8B-128k and LLaMA3-8B-128k, using scaling factors from two PPL-guided searches. The results show that naive PPL-guided search fails to ensure effective rescaling factors, as it struggles to identify the correct critical dimension and tends to yield slightly smaller scaling factors.

**The effectiveness of mixed context window training**. To ablate its effectiveness, we disable mixed context window training in LongRoPE2 and instead follow conventional mid-

ample, YaRN and NTK show performance drop of -15.2% and -9.3% oh Phi3-mini, with declines of -21.15 and -14.55 absolute points on GSM8K. In contrast, LongRoPE2 retains 97.6% and 98.6% o0f the pre-trained performance on Phi3-mini-3.8B and LLaMA3-8B, establishing it as the first lossless extension method that preserves core capabilities.

## 4.3 Ablation Study

**The effectiveness of real critical dimension** $d_{rcd}$. A key factor in LongRoPE2's superior long-context performance is its full resolution of RoPE OOD values across all dimensions. To validate this, we extend our experiments beyond LongRoPE2 by applying our identified practical critical dimension $d_{rcd}$ to YaRN and NTK, yielding YaRN-rcd and

training with a single rescaled RoPE. As shown in Table 7, removing mixed context window training results in a significant drop in performance on regular short-context tasks, as expected. Interestingly, mixed context window training not only preserves short performance but also improves long-context performance (8k–128k). This may be attributed to the preservation of pre-trained RoPE for shorter contexts, allowing long-context training to focus more effectively on adapting to the new introduced token positions.

## 5   Related Works

In addition to methods based on RoPE rescaling, this section discusses related works of other approaches.

**RAG and Agent-based extension**. Retrieval-Augmented Generation (RAG) approaches incorporate an external memory module to store and manage long past context, coupled with dynamic retrieval mechanisms to fetch task-relevant documents during inference (Jeong et al., 2024; Chan et al., 2024; Dong et al., 2024; Gutiérrez et al., 2025; Luo et al., 2024). Agent-based methods, meanwhile, decompose long-context processing into iterative planning, summarization, and retrieval tasks, often employing multi-agent workflows: individual agents extract information from text segments, which are aggregated to bypass fixed context limits (Zhang et al., 2024b; Li et al., 2024b; Lee et al., 2024b), while others integrate specialized architectures (e.g., hierarchical attention) for direct long-text handling (Gur et al., 2024). Both directions—relying on external modules or multi-step decomposition—are complementary to our method.

**Efficient long-context modeling**. Attention computation and memory costs grow quadratically with context length, prompting research into reducing these challenges through improved attention mechanisms and innovative model structures. Many methods leverage the sparsity of standard attention, reducing computation by focusing on local and auxiliary regions (Child et al., 2019; Beltagy et al., 2020; Zaheer et al., 2020; Guo et al., 2022), while others extend context length using fine-grained sparsity (Ding et al., 2023) or chunked attention (An et al., 2024). Linear attention approaches further lower complexity while achieving comparable performance, with additional optimization for hardware efficiency (Katharopoulos et al., 2020; Yang et al., 2024c). State-space models (SSMs) offer linear complexity for sequence modeling (Gu & Dao, 2024; Yu et al., 2024), and hybrid transformer-SSM architectures enhance foundational model capabilities (Lieber et al., 2024; Ren et al., 2024). Most of these approaches build upon RoPE, making them complementary to our approach.

## 6   Conclusion

We present LongRoPE2, a method for near-lossless LLM context window extension. By addressing insufficient train-

ing of higher RoPE dimensions—a key limitation in handling OOD positional values—LongRoPE2 uses evolutionary search-guided rescaling and mixed context window training to achieve 128k effective context length with just 10B tokens, retaining 97.6% of the original short-context performance. Extensive experiments on on LLaMA3-8B and Phi3-mini-3.8B demonstrates the superiority over prior art approaches. Future work will explore scaling LongRoPE2 toward fully lossless and infinite context window extension.

## Impact Statement

This work advances the field of Machine Learning by enabling LLMs to process longer contexts effectively. LongRoPE2 enhances LLM capabilities for tasks like document summarization and scientific research. There are many potential societal consequences of our work, none of which we feel must be specifically highlighted here.

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

# A    Additional Experiments and Analysis

**Additional details**. For the rescaling factor search, we set a population size of $P = 64$, evolution iterations of 40, and a mutation probability $p = 0.3$. The searched rescaling factors are then applied with mixed context window training.

To accelerate training and inference, we use FlashAttention-2 (Dao, 2023), which requires no modifications for mixed context window training or factor-switch-based inference (as illustrated in Fig. 10). Given that GPU memory and computation time increase exponentially with sequence length, fine-tuning long-context models presents significant challenges. To address this, we utilize nnScaler (Lin et al., 2024), an efficient distributed training system for long-context LLMs, to reduce training costs. 10B tokens take approximately 39 hours for Phi3-mini and 54 hours for LLaMA3-8B on 64 A100 GPUs. During inference, the switch between rescaled and original RoPE is triggered when the combined length of the input context and generated tokens exceeds the pre-trained context window. Switching to rescaled RoPE for long-context inference requires a one-time recalculation of the KV cache, a potential limitation we leave for future work.

**Additional results on RULER and Needle-in-a-Haystack**. Tables 8 and 9 show the detailed per-task accuracy of our extended LLMs on the RULER benchmark. Figures 7 and 8 provide comprehensive results for the needle-in-a-haystack tests. As observed, the YaRN method frequently fails to retrieve needles across Phi3-mini-3.8B, LLaMA3-8B, Meta-LLaMA3.1-8B and Meta-LLaMA3.1-8B-Instruct.

*Table 8.* LongRoPE2-extended Phi3-mini (3.8B)-128k per-task performance on RULER.

| Length | NIAH single1 | NIAH single2 | NIAH single3 | NIAH multikey1 | NIAH multikey2 | NIAH multikey3 | NIAH multivalue | NIAH multiquery | VT | CWE | FEW | single-hop QA | multi-hop QA | Avg. |
|---|---|---|---|---|---|---|---|---|---|---|---|---|---|---|
| 4096 | 100 | 100 | 99 | 91 | 96 | 97 | 97.75 | 97.75 | 85.8 | 93.7 | 85.33 | 82 | 50 | 90.41 |
| 8192 | 100 | 100 | 100 | 90 | 93 | 97 | 89.5 | 93.75 | 84 | 87.2 | 86 | 68 | 47 | 87.34 |
| 16384 | 100 | 100 | 99 | 87 | 88 | 82 | 91.25 | 89 | 85 | 55.4 | 91.67 | 70 | 45 | 83.33 |
| 32768 | 100 | 100 | 99 | 86 | 86 | 57 | 87 | 78 | 76.8 | 33.2 | 91.67 | 56 | 44 | 76.51 |
| 65536 | 100 | 100 | 99 | 85 | 71 | 32 | 67.75 | 69.25 | 66.8 | 0.4 | 71.67 | 50 | 37 | 65.37 |
| 131072 | 100 | 98 | 95 | 92 | 40 | 18 | 56.75 | 59 | 35.2 | 0.3 | 89.33 | 47 | 34 | 58.81 |

*Table 9.* LongRoPE2-extended LLaMA3-8B-128k per-task performance on RULER.

| Length | NIAH single1 | NIAH single2 | NIAH single3 | NIAH multikey1 | NIAH multikey2 | NIAH multikey3 | NIAH multivalue | NIAH multiquery | VT | CWE | FEW | single-hop QA | multi-hop QA | Avg. |
|---|---|---|---|---|---|---|---|---|---|---|---|---|---|---|
| 4096 | 100 | 100 | 99 | 100 | 100 | 100 | 99 | 99.75 | 98.8 | 98.5 | 96.33 | 79 | 60 | 94.61 |
| 8192 | 100 | 100 | 100 | 100 | 100 | 100 | 99 | 99.75 | 99.8 | 95.9 | 91.33 | 74 | 58 | 93.68 |
| 16384 | 100 | 100 | 100 | 99 | 100 | 98 | 95 | 98.25 | 99.6 | 86.8 | 96.33 | 69 | 58 | 92.31 |
| 32768 | 100 | 100 | 100 | 99 | 98 | 100 | 98 | 96.25 | 98.6 | 63.9 | 95.67 | 72 | 55 | 90.49 |
| 65536 | 100 | 100 | 100 | 98 | 98 | 95 | 95.75 | 99.75 | 98.6 | 33.6 | 80.33 | 62 | 52 | 85.62 |
| 131072 | 100 | 100 | 99 | 96 | 91 | 94 | 96.5 | 97 | 92.6 | 9 | 85.33 | 56 | 50 | 82.03 |

*Table 10.* Ablation study on the number of searched dimensions.

| Method | Regular short tasks | | | RULER | | | | | |
|---|---|---|---|---|---|---|---|---|---|
| | MMLU | MMLU Pro | GSM8K | 4k | 8k | 16k | 32k | 64k | 128k |
| Base Model: Phi3-mini (3.8B) | | | | | | | | | |
| LongRoPE2 ($d_{rcd}$ and higher dims) | **70.07** | **40.30** | 73.62 | **90.41** | **87.22** | **83.33** | **76.51** | **65.37** | **58.81** |
| LongRoPE2 (all dims) | 69.96 | 39.84 | **74.83** | 90.02 | 87.21 | 82.42 | 74.86 | 63.95 | 57.34 |
| Base Model: LLaMA3-8B | | | | | | | | | |
| LongRoPE2 ($d_{rcd}$ and higher dims) | **65.01** | **34.61** | 50.80 | **94.61** | **93.68** | **92.31** | **90.49** | **85.62** | **82.03** |
| LongRoPE2 (all dims) | 64.34 | 33.83 | **51.55** | 93.92 | 92.61 | 91.41 | 89.30 | 83.11 | 78.07 |

**The ablation study on search algorithm**. In our work, we focus on searching for the real critical dimension and the scaling factors of higher dimensions beyond it. For the lower dimensions before the critical dimension, we directly apply NTK scaling without further optimization. To evaluate this design, we conduct an additional ablation study. For comparison, we also allowed the search to include lower dimensions. As shown in Table 10, while searching across all dimensions yields competitive results, it underperforms compared to our proposed method. A possible reason is that limiting the search to higher dimensions significantly reduces the search space, enabling a more effective discovery of the optimal solution.

**Discussions on long-short RoPE factor switch**. When transitioning from the short context window (which uses the short factor, i.e., the original RoPE) to the long context window (which uses the rescaled long factor), KV cache recomputation is

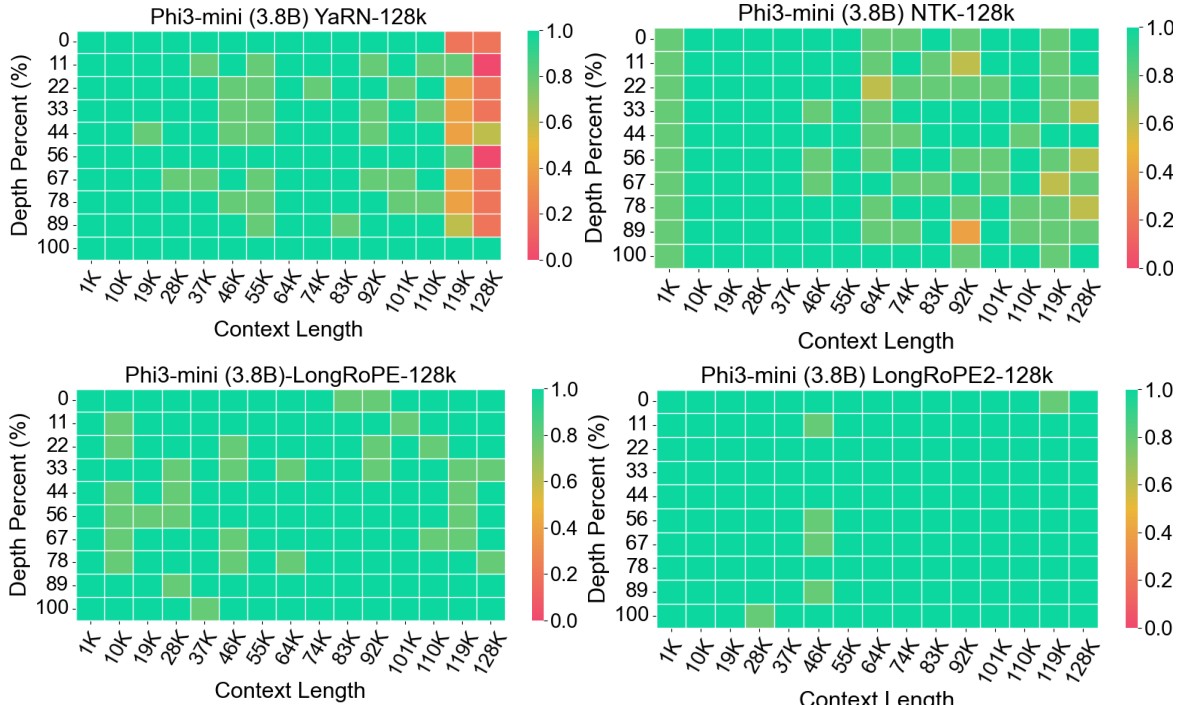

*Figure 7.* Needle in a Haystack full results for Phi3-mini (3.8B)-128k.

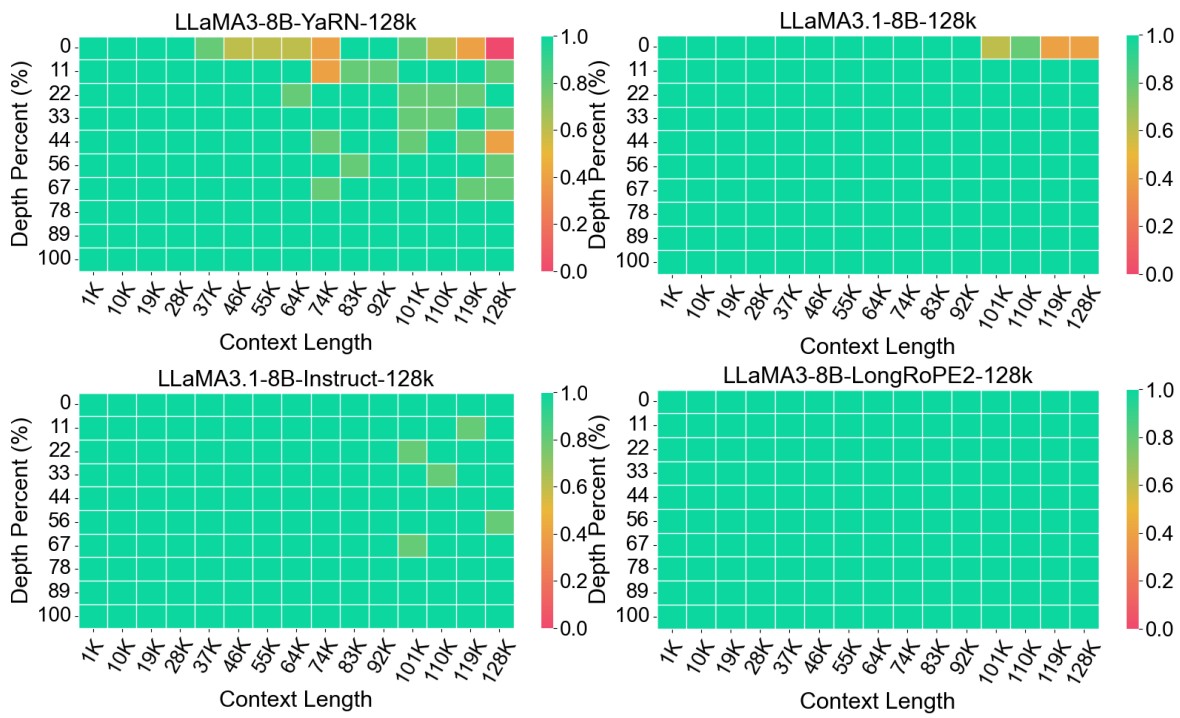

*Figure 8.* Needle in a Haystack full results for LLaMA3-8B-128k.

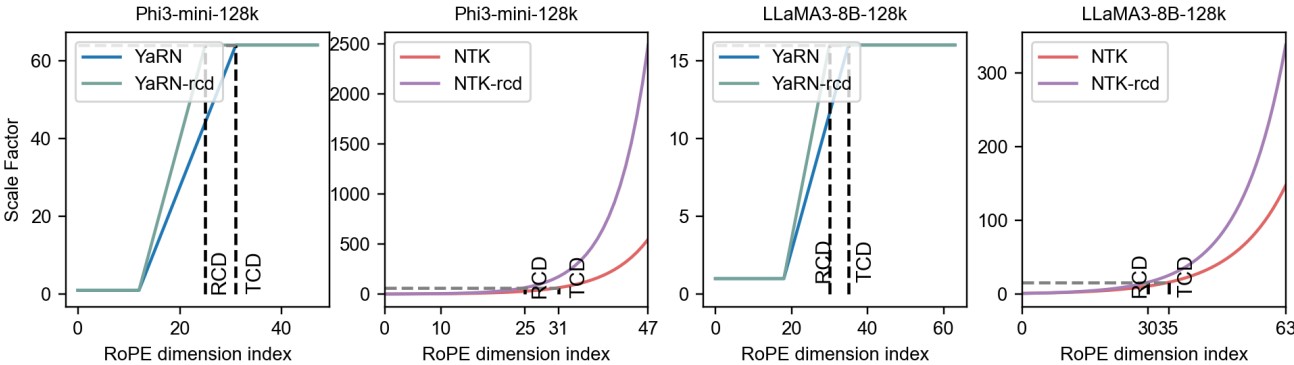

*Figure 9.* The RoPE rescaling factor distributions of NTK/YaRN adjusted based on the real critical dimension (i.e., YaRN-rcd, NTK-rcd).

```
# Data: <BOS>[TEXT1]<EOS>...<BOS>[TEXTN]<EOS>
# flash attention api call in pre-training
from flash_attn import flash_attn_func

attn_output = flash_attn_func(
            query_states,
            key_states,
            value_states,
            dropout_p=dropout,
            causal=True)

# flash attention api call in mixed context window training
from flash_attn import flash_attn_varlen_func
attn_output = flash_attn_varlen_func(
            query_states,
            key_states,
            value_states,
            dropout_p=dropout,
            causal=True,
            cu_seqlens_q=cu_seq_lens_q,
            cu_seqlens_k=cu_seq_lens_k,
            max_seqlen_q=max_length_q,
            max_seqlen_k=max_length_k)
```

```
def origin_rope(position_ids, base, rope_dim):
    ...
    inv_freq = 1.0 / (base**torch.Tensor([0, 1, 2, 3, ..., rope_dim]))
    ...
    return cos, sin

def longrope(position_ids, base, rope_dim, original_max_position_embeddings,
long_factor, short_factor):
    ...
    ori_inv_freq = 1.0 / (base**torch.Tensor([0, 1, 2, 3, ..., rope_dim]))

    # switch to rescaled RoPE if the current seq length exceeds the original context window
    if (torch.max(position_ids) + 1) > original_max_position_embeddings:
        ext_factors = long_factor
    else:
        ext_factors = short_factor # use the original RoPE
    inv_freq = ori_inv_freq / ext_factors
    ...
    return cos, sin
```

*Figure 10.* The pseudocode for mixed context window training and inference.

required. However, this recomputation does not occur during every inference. It is triggered only when the input length remains within the short context window, but the total sequence length (including both input and generated tokens) exceeds it for the first time. After this one-time recomputation, no further recomputation is needed for the remainder of the generation. In practice, this situation is relatively uncommon, as most inference cases either remain within the short context window or begin directly in long-context mode.

To quantify the computational overhead, we measured the KV cache recomputation time on a 4×80GB A100 GPU (using vLLM 0.7.3) for both Phi-3-mini and LLaMA3-8B, and compared it against normal decoding latency. As shown in Table 11, the additional recomputation cost is equivalent to generating only 15 (phi3-mini) and 25(llama3-8b) tokens, which is negligible in the context of long-context generation.

*Table 11.* Prefil and decoding time costs. The numbers in () indicate the amount of decoded tokens corresponding to the time spent on KV cache recomputation.

| Model | Prefil time (KV recompute) | Decode time across different generation length | | | | | |
|---|---|---|---|---|---|---|---|
| | | 512 | 1k | 2k | 4k | 8k | 16k |
| Phi3-mini (prefill 2k) | 124.1ms | 7.6 ms (*16.2*) | 7.66 ms (*16.2*) | 7.7 ms (*16.1*) | 7.8 ms (*15.9*) | 14.3 ms (*8.7*) | 23.3 ms (*5.3*) |
| LLaMA3-8B (prefill 8k) | 613.9 ms | 24.1 ms (*25.5*) | 24.2 ms (*25.3*) | 24.1 ms (*25.5*) | 24.2 ms (*25.4*) | 23.5 ms (*26.1*) | 23.6ms (*26.0*) |

# B  Synthetic data sample

---

**Synthetic search data based on a PG19 book sample**

A special magic number is hidden within the following text. Make sure to memorize it. I will quiz you about the number afterwards.

One of the special magic numbers for numerous-kite is: 6716097. The Old Testament of the King James Version of the Bible The First Book of Moses: Called Genesis 1:1 In the beginning God created the heaven and the earth. 1:2 And the earth was without form, and void; and darkness was upon the face of the deep. And the Spirit of God moved upon the face of the waters.

......

it be for a witness between me and thee. 31:45 And Jacob took a stone, and set it up for a pillar. 31:46 And Jacob said unto his brethren, Gather stones; and they took stones, and made an heap: and they did eat there upon the heap. 31:47 And Laban called it Jegarsahadutha: but Jacob called it Galeed.

......

3:39 Also in the fifteenth day of the seventh month, when ye have gathered in the fruit of the land, ye shall keep a feast unto the LORD seven days: on the first day shall be a sabbath, and on the eighth day shall be a sabbath. 23:40 And ye shall take you on the first day the boughs of goodly trees, branches of palm trees, and the boughs of thick trees, and willows of the brook; and ye shall rejoice before the LORD your God seven days. What is the special magic number for numerous-kite mentioned in the provided text? The special magic number for numerous-kite mentioned in the provided text is

---

