# OpenReview forum: "LongRoPE2: Near-Lossless LLM Context Window Scaling"
_ICML.cc/2025/Conference — ICML 2025 poster_

### Official Review · Reviewer_ak4t · 2025-03-12

**Overall Recommendation:** 3

**Summary:**

This paper mainly introduces LongRoPE2, aiming to achieve an effective long context window while preserving short-context performance by context extension. Based on LongRoPE, LongRoPE2 introduces a new needle-PPL guided evolutionary search method for settling the rescaling factors, and proves it to be more effective than the naive PPL-guided one by experiments. For retaining preformance on short contexts, LongRoPE2 proposes a novel mixed context window training method. Compared to YaRN, NTK and LongRoPE,  LongRoPE2 achieves better performance on long contexts while retaining over 98.5% of short-context performance.

---

## update after rebuttal

Thanks to the authors for their response, and I acknowledge that LongRoPE2 is a strong and valuable work. However, there are still some unclear aspects in the paper, such as the lack of a detailed explanation regarding how Figure 3(a) was derived. Due to these unresolved concerns, I have decided not to adjust our initial score. In our view, a score of 3 (borderline accept) remains reasonable and justified.

**Claims And Evidence:**

yes

**Essential References Not Discussed:**

No

**Experimental Designs Or Analyses:**

Yes.
In 4.2, this paper presents reults on RULER, NIAH, LOFT of LongRoPE2-extened, comparing with other SOTA RoPE rescaling methods(YaRN, NTK, LongRoPE) and shows the effectiveness.
In 4.3,  to validate the effectiveness of real critical dimension, one experiment applies d_{rcd} to YaRN and NTK, which is also get improved.  To validate the effectiveness of needle-PPL guided search, one experiment compares it with the naïve one with the same training process on the same test dataset. Finally, the effectiveness of mixed context window training is also validated here.

**Methods And Evaluation Criteria:**

YES

**Other Comments Or Suggestions:**

There is one typo on page 8, subtitle: need-PPL  should be needle-PPL

**Other Strengths And Weaknesses:**

This paper is clearly demonstrated. The most enlightening contribution may be the New RoPE OOD hypothesis (in 3.1), as it gives directions for optimization of other methods  not limited to this paper.

**Questions For Authors:**

No

**Relation To Broader Scientific Literature:**

Some scaling methods before LongRoPE(PI, YarN, NTK) ignore the actual errors caused by different parameters of models after training. This issue was preliminarily solved in LongRoPE. LongRoPE2 is based on LongRoPE, and more effective.

**Theoretical Claims:**

yes
This paper proposes a New RoPE OOD Hypothesis that the empirical RoPE periods in higher dimensions are longer than theoretical values, limiting current methods to fully address RoPE OOD. This implies that the actual optimal rescaling factors may be greater than the theoretical one. Then, by applying needle-PPL-guided search, LongRoPE2 does get rescaling factors larger than the theoretical one, and performs better, which from this point of view can test this hypothesis.

---

> ### Author Rebuttal · Authors · 2025-03-30
>
> Thank you for your thoughtful review and for recognizing our contributions. We greatly appreciate your acknowledgment of our New RoPE OOD Hypothesis and the role of needle-PPL-guided search in validating this hypothesis through empirical results. We are also glad that you found our extensive experiments in Sections 4.2 and 4.3 valuable in demonstrating the effectiveness of LongRoPE2 and our key design choices, such as the real critical dimension and mixed context window training. Please let us know if there are any specific aspects where we can provide more details. Thank you again for your time and constructive evaluation!

---

> > ### Comment · Reviewer_ak4t · 2025-04-02
> >
> > Thanks for your response. I have no more questions and will keep my score.

---

### Official Review · Reviewer_gztC · 2025-03-13

**Overall Recommendation:** 3

**Summary:**

This paper proposed LongRoPE2, a RoPE scaling method to extend the context window of LLMs. The primary extension compared to LongRoPE1 is that LongRoPE2 utilizes a needle-based search rather than perplexity-based one for various rope dimension scaling. The experimental results demonstrate the superior performance of LongRoPE2 compared to other RoPE scaling methods.

**Claims And Evidence:**

1. The most overclaim is  "LongRoPE2-extended LLaMA3-8B-128k surpasses Meta’s LLaMA3.1-8B-128k in long-context performance with 80x fewer training tokens". This claim is supported by the RULER results in Fig 1. However, LongRoPE adopts a needle-based search for RoPE scaling, which may (over)fit the synthetic tasks in RULER benchmark, hence achieving better results. For other general tasks such as En. MC in InfiniteBench, **LongRoPE2-LLaMA3-8B achieved a score of 46.72 but LLaMA-3.1-8B achieved 65.1**. This means LongRoPE2-LLaMA3-8B may still have a large gap to LLaMA-3.1-8B involving far more training tokens. Note that it's not necessary for LongRoPE2-LLaMA3-8B to surpass LLaMA-3.1-8B, but should fix the claim for clearness.

2.  The mixed context window training is adopted in [1] and LLaMA-3.1 (as well as a common practise in long-context LLM community) to maintain the short-context performance, but the authors claims they propose such a "novel" strategy.

[1]LongAlign: A Recipe for Long Context Alignment of Large Language Models

**Essential References Not Discussed:**

This paper considers a mix context window training strategy as one of its primary contributions. However, such strategies are widely-used in some previous works such as LongAlign[1], which is missing in references.

[1]LongAlign: A Recipe for Long Context Alignment of Large Language Models

**Experimental Designs Or Analyses:**

I have gone through the ablation studies which have demonstrated the effectiveness of LongRoPE2's designs.

**Methods And Evaluation Criteria:**

The evaluation benchmarks are popular in the long-context understanding field.

**Other Comments Or Suggestions:**

The related work section is better placed in the main body of the paper to make it self-contained. (This is a suggestion and wouldn't affect my rating.)

**Other Strengths And Weaknesses:**

Strength:
1. Generally the experiments covering long and short contexts are well-designed and can demonstrate the effectiveness of the proposed method.
2. The intuition regarding the insufficient training of high-frequencies RoPE makes sense.
3. I believe the needle-driven PPL search is a better choice for LongRoPE as the pure PPL on normal documents may be orthogonal to long-context performance.

Weaknesses:
1. I feel most designs in the work have been proposed/adopted in previous works. For example, the intuition that high-frequencies RoPE  may be insufficiently trained has been introduced in a popular blog (https://spaces.ac.cn/archives/9706) regarding RoPE scaling. The mixed context window training is adopted in [1] and LLaMA-3.1 to maintain the short-context performance. It may be improper to regard these points as this work's contributions and should give some credit to the related works.

[1]LongAlign: A Recipe for Long Context Alignment of Large Language Models

**Questions For Authors:**

1. Do you think it's better to list the results of LLaMA-3.1-8B in the main table? Since you claimed the proposed method on 10B tokens can surpass LLaMA-3.1-8B's continual training of 800B tokens.

2. What are the criteria for you to choose the evaluation tasks? It seems there are some challenging tasks such as En.QA, En. Sum, etc. in the InfiniteBench and some other tasks of various categories in LongBench. The selected tasks seem to be irregular.

3. The claim regarding the intuition of insufficient high-frequencies RoPE training and the mix context window training should be better to fix.

**Relation To Broader Scientific Literature:**

This paper is a direct extension of LongRoPE[1].

[1] LongRoPE: Extending LLM Context Window Beyond 2 Million Tokens

**Theoretical Claims:**

There are no proofs for theoretical claims.

---

> ### Author Rebuttal · Authors · 2025-03-30
>
> >Q1: Clarification on LLaMA3.1-8B long-context evaluation numbers, and the "overclaim" comments
>
> **Response**: We appreciate your feedback and would like to clarify the following points:
>
> 1. **65.1 is the En.MC score of the instruct version, not LLaMA3.1-8B.**:  As noted in Table 2 of the LLaMA3.1 tech report, the 65.1 score is for the instruct-tuned version (a detail that can be overlooked). Compared to the fair baseline, LLaMA3.1-8B, our model achieves a higher score (**46.72** vs. **45.85**) on En. MC. Moreover, our model consistently outperforms LLaMA3.1-8B across several long-context benchmarks. Here are additional results:
>
> >InfiniteBench and LongBench:
>
> ||avg.|En.MC|En.Sum| KV retrieval | TriviaQA | TREC | LCC | RepoBench-P|
> |:--:   |:--:   |:--:   |:--:   |:--:   |:--:   |:--:   |:--:   |:--:   |
> |LLaMA3.1-8B|54.28| 45.85| 15.27|16.20| **91.13**| 73.50| 70.24| **67.83**|
> |LongRoPE2-LLaMA3-8B|**65.20**|**46.72**| **16.20**|**88.0** |**91.13**| **76.50**| **70.47**| 67.39 |
>
> >LOFT:
>
> ||avg.| ArguAna | FEVER | HotPotQA | MS MARCO | NQ | Quora | SciFact|
> | :--:  | :--:  | :--:   | :--:   |:--:   |:--:   |:--:   |:--:   |:--:   |
> |LLaMA3.1-8B| 53.14|19.0| 90.0| 12.0| 69.0| 78.0| 61.0|43.0|
> |LongRoPE2-LLaMA3-8B| **74.28**|**28.0** | **96.0** | **70.0** | **80.0**| **94.0**| **79.0** | **73.0**|
>
>
> 2. **Our RoPE scaling method is designed to improve broad long-context capabilities, not to optimize for any specific benchmark like RULER.** The use of needle data for search is **not** designed to fit RULER but to **better control long-range token dependency distances** in long documents.  E.g., We used only the simplest number needle synthesis method. Our extensive experiments proved the superiority over other methods (e.g., NTK, YaRN) across diverse benchmarks.
>
> >Q2: What are the criteria for you to choose the evaluation tasks?
>
> **Response**: Our selection follows two key principles:
> 1) **Effectiveness for evaluating a pre-trained LLM rather than a chat LLM**. Since our method extends a pre-trained LLM without post-training, we prioritize tasks aligned with this setup: (i) completion-based tasks, such as few-shot learning and code completion and En.MC in InfiniteBench. (ii) QA tasks with few-shot examples, such as various text-retrieval QA tasks in LOFT.
>
> 2) **Comprehensive long-context evaluation**. To evaluate multiple aspects of long-context performance, we include tasks covering RULER, needle-in-a-haystack retrieval, real-world text QA, high-difficulty KV retrieval, multi-choice QA, few-shot learning, and code completion, as detailed in our evaluation section. We believe this selection fairly reflects the strengths of our method and provides a well-rounded assessment.
>
> *Additional results on chat-based sub-tasks*.  For your reference, we provide additional results on chat-based LongBench tasks. As shown below, We achieve the highest average score, even surpassing LLaMA3.1-8B.
>
> ||Avg.|narrativeqa| Qasper | multifiledQA | hotpotqa | 2wikimqa | musique | gov_report | qmsum| samsum|
> | :--: | :--:  | :--:  | :--:   | :--:   |:--:   |:--:   |:--:   |:--:   |:--:   |:--:   |
> |LLaMA3.1-8B|22.60| 20.90|12.50|32.72|11.95| 13.98| **8.62**| 29.95|**25.53**|**47.23**|
> |NTK-LLaMA3-8B|20.32|21.14|11.93|29.02|11.91|14.71|7.81|21.50|22.09|42.70|
> |LongRoPE2-LLaMA3-8B|**24.31**|**21.79**|**18.13**|**36.25**|**13.85**|**19.42**|8.03|**30.12**|25.41|45.80|
>
> >Q3: The clarifications on the main contributions and claims:
>
> **Response**: We are grateful for your questions and would like to clarify that our main contribution is not the discovery of insufficient training in high-frequency RoPE, but rather the introduction of **a new RoPE OOD hypothesis**. This hypothesis explains why existing RoPE rescaling methods, such as NTK and YaRN, often result in suboptimal long-context performance. This contribution has been acknowledged by the other two reviewers.
>
> Regarding the mixed context window training, we would like to emphasize that the key difference between our approach and those used in LLaMA3.1 and LongAlign is the use of two distinct RoPE scaling factors: a short factor for short contexts and a long factor for long contexts. This **dual-factor** approach is essential in significantly recovering short-context performance, which we have shown through extensive experiments.
>
> Here, we perform an additional comparison with LLaMA-3.1’s mixed training. As shown below, we significantly improve short-text performance.
>
> ||(short)-MMLU|(short)-MMLU pro| (short)-GSM8k | Ruler-128k|
> | :--: | :--:  | :--:  | :--:   | :--:   |
> |LLaMA3-8B (**Our mixed context windows training**)|**65.01**|**34.61**|**50.80**|**82.03**|
> |LLaMA3-8B (mixed training in LLaMA3.1)| 64.18|32.95 |46.25 |71.83 |
>
> We hope these responses address your concerns and clarify any confusion, and we will incorporate them in the revisions. Thank you again for your valuable feedback and suggestions, and we kindly ask you to consider re-evaluating our work.

---

> > ### Comment · Reviewer_gztC · 2025-04-02
> >
> > 1. Thanks for addressing my confusion and now I believe the effectiveness compared to LLaMA-3.1. But I feel the infinitebench is quite unstable, i.e., the performance before and after instruction tuning will be quite distinct, though there are no requirements to follow specific formats. I hope to see more results on some new benchmarks such as LongBench v2 in the future.
> >
> > 2. The mix context window training still seems to be similar to previous works. The only difference is to adjust RoPE factor for different lengths, which is also used in a previous work[1]. I feel it is better to be a training strategy inspired by previous works rather than the main contribution of this paper.
> >
> > [1] CLEX: Continuous Length Extrapolation for Large Language Models
> >
> > Other concerns have been addressed. Now I can raise my score to 3.

---

> > > ### Author Response · Authors · 2025-04-02
> > >
> > > Thank you for your thoughtful follow-up. We're glad our response clarified the effectiveness of our method compared to LLaMA-3.1.
> > >
> > > We appreciate your suggestions. We agree that evaluating on newer benchmarks, such as LongBench v2, would be valuable and will consider this in future work. Regarding mixed context window training, we will refine the discussion in our revision to clarify its position more clearly.
> > >
> > > Thanks again for your valuable feedback!

---

### Official Review · Reviewer_Ebnv · 2025-03-14

**Overall Recommendation:** 4

**Summary:**

Maintaining the performance on both long and short benchmarks are a critical challenge for existing long context extension methods. LongRoPE2 is a new approach that extends the effective context window of pre-trained large language models to the target length, while preserving the performance on the original shorter context window.

**Claims And Evidence:**

Claims: LongRoPE2 extends context windows to 128k while retaining >97% short-context performance. The key contributions are (1) higher RoPE dimensions are undertrained, (2) evolutionary search for rescaling factors guided by needle-driven perplexity, (3) mixed training with original/rescaled RoPE.

Evidence: Achieves strong results on RULER, and real-world benchmarks (LOFT, LongBench). Outperforms YaRN, NTK, and LongRoPE with much fewer tokens.

**Essential References Not Discussed:**

No

**Experimental Designs Or Analyses:**

Ablations confirm needle-PPL’s superiority over standard PPL and mixed training’s necessity.
Adjusted baselines (YaRN-red/NTK-red) show improved but suboptimal performance.

**Methods And Evaluation Criteria:**

Methods: Evolutionary search for critical dimensions and scaling factors, mixed training (original RoPE for short contexts, rescaled RoPE for long).

Evaluation: Benchmarked on RULER, Needle-in-a-Haystack (retrieval), LOFT/InfiniteBench (real-world), and MMLU/GSM8K (short-context).

**Other Comments Or Suggestions:**

N/A

**Other Strengths And Weaknesses:**

Strengths: Efficient scaling (10B tokens), minimal short-context degradation.

Weaknesses: Evolutionary search computational cost; inference requires KV cache recalculation.

**Questions For Authors:**

1. How does evolutionary search scale to million-token contexts?
2. Does mixed training cause interference between short/long contexts?

**Relation To Broader Scientific Literature:**

Builds on RoPE rescaling (NTK, YaRN) and evolutionary optimization. Different from RAG/agent-based methods, positioning LongRoPE2 as complementary methods.

**Theoretical Claims:**

Challenges prior RoPE OOD theory: insufficient training in higher dimensions extends empirical periods, requiring larger scaling factors than theoretical bounds.

To be honest I do not carefully check the correctness of all theoretical claims of this paper.

---

> ### Author Rebuttal · Authors · 2025-03-30
>
> **Response**: Thank you for your valuable feedback and for recognizing the strengths of our work. We appreciate the opportunity to address your concerns.
>
> 1) **Affordable evolutionary search computational cost**: we acknowledge that evolutionary search introduces additional costs. To further clarify its feasibility, we conduct additional experiments to evaluate the search cost when scaling from 128k (current context window length) to 1024k. Using vllm0.7.3 as the inference engine and running on an 8*A100(80GB) server setup, we measured the total search time. As shown in the table below, even when scaling to 1M tokens, the total search time remains manageable at 240 hours (10 days). Moreover, this is a one-time *offline* process, and the search time can be linearly reduced by increasing the number of GPUs, due to the nature of evolutionary search. Therefore, it is practical for LLM pretraining teams.
>
> ||128k|512k| 1024k|
> |:--:   |:--:   |:--:   |:--:   |
> |total search time on 8*80GB a100| 7.5h | 68h | 240h |
>
> 2) **KV cache recalculation occurs only in specific cases and has minimal overhead**: We acknowledge that KV cache recomputation is required when transitioning from the short context window (using the short factor, i.e., original RoPE) to the long context window (using the long factor). However, this recomputation does not occur in every inference. It happens only when the input length is within the short context window, but the total length (input+generated tokens) exceeds it **for the first time**. After this one-time recomputation, no further recomputation is needed for the rest of the generation.
>
> In most general inference scenarios, this situation is relatively uncommon, as prompts and completions typically either remain within the short context window or start in the long context mode from the beginning. To quantify the cost, we measured KV recomputation time on a  4x80GB A100 GPU (with vllm 0.7.3) for Phi-3-mini and LLaMA3-8B, comparing it against normal decoding time:
>
> ||prefill (kv recompute)|decode-output 512| decode-output 1k| decodetime-output 2k| decodetime-output 4k| decodetime-output 8k | decodetime 16k|
> |:--:   |:--:   |:--:   |:--:   |:--:   |:--:   |:--:   |:--:   |
> |Phi3-mini (prefill 2k)| 124.1ms| 7.63ms (**16.2**)| 7.66ms (**16.2**)| 7.71ms (**16.1**)| 7.78ms (**15.9**)| 14.29ms (**8.7**)| 23.3ms (**5.3**) |
> |LLaMA3-8B (prefill 8k)|613.9ms | 24.11ms (**25.5**)| 24.22ms (**25.3**)| 24.05ms (**25.5**)| 24.18ms (**25.4**) | 23.5ms (**26.1**) | 23.58ms (**26.0**) |
>
> The numbers in () indicate the amount of decoded tokens corresponding to the time spent on KV cache recomputation. These results indicate that the additional recomputation cost is equivalent to generating only ~15 (phi3-mini) and ~25(llama3-8b)  tokens, which is negligible in the context of long-context generation.
>
>
> >Q2: Does mixed training cause interference between short/long contexts?
>
> **Response**: Thank you for your insightful question. While it’s true that mixed context window training applies two RoPE scaling factors simultaneously during mid-training, which could introduce interference, our empirical results suggest that this "interference" plays a constructive role and hence does not degrade performance. In fact, it not only recovers short-context performance but also enhances long-context performance. To better illustrate this, we refer to Table 7 from our original paper.
>
> ||MMLU-(short)|MMLUPro-(short)|GSM8K-(short)| RULER-4k| RULER-8k| RULER-16k | RULER-32k|RULER-64k|RULER-128k|
> |:--:   |:--:   |:--:   |:--:   |:--:   |:--:   |:--:   |:--:   |:--:   |:--:   |
> |Phi3-mini (**with mixed context window training**)| **70.07**|**40.30**|**73.62**|90.41|**86.87**|**83.33**|**76.51**|**65.37**|**58.81**|
> |Phi3-mini (no mixed context window training)|66.56|34.86|64.67|**90.55**|85.77|81.08|73.31|63.75|56.22|
> ||||||||||
> |LLaMA3-8B (**with mixed context window training**)|**65.01**|**34.61**|**50.80**|94.61|**93.68**|**92.31**|**90.49**|**85.62**|**82.03**|
> |LLaMA3-8B (no mixed context window training)| 64.57| 33.83| 48.37| **94.67**|93.15|91.24|89.38|83.53|80.18|
>
> A possible explanation for this surprising improvement is that the so-called “interference” actually plays a constructive role in learning. Specifically, the short-context window helps preserve position modeling for non-interpolated positions (e.g., LLaMA3’s native positions 0, 1, 2, ..., 8191), while the long-context window primarily facilitates the adaptation for newly interpolated positions (e.g., LLaMA3’s new positions like 1/16, 2/16, ..., 17/16). This training strategy effectively constrains the model’s adaptation to interpolated positions while maintaining consistency with the original position modeling - a concept similar to the KL divergence constraint in PPO, which prevents large deviation from the original policy model.
>
> We appreciate this insightful question, which has prompted further reflection and discussions.

---

### Decision · Program_Chairs · 2025-05-01

**Decision:**

Accept (poster)

**Comment:**

The study presents LongRoPE2, a method designed to significantly extend the context window of pre-trained large language models, such as LLaMA3-8B and Phi3-mini-3.8B, without compromising the performance on shorter context windows. By addressing the out-of-distribution (OOD) issues with existing methods, LongRoPE2 employs a hypothesis-driven evolutionary search algorithm to optimize the RoPE (Random Projections) dimensions, and introduces a mixed training approach. combines original and rescaled RoPE to ensure high performance across various benchmarks. The paper highlights extensive experiments that validate the effectiveness of LongRoPE2, demonstrating its ability to extend the context window to 128k while retaining over 98.5% of the short-context performance using only 10B tokens, a substantial improvement over previous approaches.
Reviewers raised concerns including affordable evolutionary search computational cost, KV cache recalculation, needle-PPL and mixed training necessity，evaluation tasks, effectiveness of the method, etc.  The authors addressed these concerns with clarifies and additional experimental results.
The method has the potential to significantly impact the field of large language models by enabling longer context windows, which is crucial for many real-world applications of LLMs.